# Using a Bed Sill as a Countermeasure for Clear-Water Scour at a Complex Pier with Inclined Columns Footed on Capped Piles

**Mahdi Esmaeili Varaki** [1,2] 🆔, **Negar Tavazo** [2] **and Alessio Radice** [3,*] 🆔

1   Department of Water and Environmental Engineering, Caspian Sea Basin Research Center,
    University of Guilan, Rasht P.O. Box 41635-1314, Iran; esmaeili@guilan.ac.ir or esmaeili.varaki@yahoo.com
2   Department of Water Engineering, University of Guilan, Rasht P.O. Box 41635-1314, Iran;
    negartavazo@gmail.com
3   Department of Civil and Environmental Engineering, Politecnico di Milano, 20133 Milan, Italy
*   Correspondence: alessio.radice@polimi.it

**Abstract:** River bridge piers may collapse due to the local scour around their foundations. It is known that local scour is an effect of the three-dimensional flow field that develops near the pier and that the geometric complexity of a non-cylindrical pier may, correspondingly, increase the complexity of the process. It is also known that various devices may be used as scour countermeasures. This manuscript explores the use of a bed sill as a countermeasure for local scour at a complex bridge pier compound of an array of piles, a pile cap, and two inclined columns with the rectangular sections above the cap. This pier geometry, never studied before in combination with a scour countermeasure, was stimulated by an existing bridge. Different sill placements were tested (at the upstream or downstream edges of the pier, or in an intermediate position) for various values of the pile diameter and number, cap thickness and cap elevation. The results of a wide experimental campaign consistently showed that the most effective placement of the transverse sill was at the upstream edge of the pier, for which scour reductions of up to 30–40% could be obtained for the long-term scour depth. The countermeasure performance decreased to about 10% when the sill was placed at the downstream edge of the pier. Furthermore, the installation of a transverse sill upstream of the pier also changed the shape of the scour hole because the pier was then located in an area prone to sill scour; however, for the present experiments, the combination of the effects was beneficial in terms of the resulting scour depth. Although the investigation of a single hydro-dynamic condition prevents the experimental findings from being generalized, the promising results stimulate further consideration of a transverse sill as a countermeasure for local scour at a complex pier.

**Keywords:** bridge scour; complex pier; counter measure; inclined columns; sill

## 1. Introduction

Bridges are key components of transportation networks, possibly interfering with river flows. One of the main causes of failure for river bridges, especially during flood times, is local scour at piers whose foundations may be undermined as the riverbed is eroded.

The flow pattern around a bridge pier includes down-flow and horseshoe vortices that cause the sediment to erode around the structures by increasing the local shear stress; furthermore, the separation of the flow at the sides of the pier creates the so-called wake vortices, which also act as tornadoes, lifting the sediment particles from the bed. As a result of these processes, a scour hole is formed around the pier and gradually increases in size until an expected equilibrium is reached (e.g., [1]). The characteristic dimensions of the vortex structures scale with the characteristic dimensions of the system (typically, the pier size and water depth), and are thus dependent on the pier geometry.

Most of the scientific literature about scour at bridge piers considers a single cylindrical pier but, for geotechnical and economic reasons, complex piers have also become popular in

bridge design, and thus attracted the interest of researchers [2]. For example, some scholars have studied the scour process at pile groups made by twin piers aligned in the stream-wise direction (e.g., [3]), or at groups with more complex configurations (e.g., [4–7]). A pile group is a relatively simple example of a complex pier, and the scour depth at a pile group is different from that around a single pile, depending on the pile spacing: for very small pile spacing, the pile group acts as a single, larger pier, and for progressively larger pile spacings, pier interference emerges and then decreases. However, complex piers typically have a more complicated shape, resulting from the combination of multiple structural elements, such as foundation piles, beams or pile caps, and columns. Each of these components may have its own effect on the scour depth. For example, Ataie-Ashtiani et al. (2010) studied the effect of pile cap elevation and pile size [5], while Ferraro et al. (2020) studied the effect of pile cap thickness on the maximum scour depth at a complex pier [8]. Yang et al. (2018, 2019) [9,10] studied the effect of a complex pier skewness on the flow and of different pier arrangements under clear-water and live-bed conditions, showing that changes in the pile-cap elevation and pier skewness could significantly change the scour depth. At a low pier skewness, the pile cap elevation was dominant and the scour depth was greatest when the pile cap was near the bed. However, as the pier was skewed to the flow, the contribution of the columns to the scour depth significantly increased. For live bed conditions, the scour pattern was significantly different, due to the upstream sediment supply, enhanced flow contraction, and reduction in the shield effect of the upstream piles to the following ones.

The columns above a pile cap may be inclined rather than vertical; for example, Figure 1 shows a picture of the 8th bridge of Ahvaz on the Karun river in Iran, which stimulated some of the present authors' research on complex piers with inclined columns. Esmaeili Varaki et al. (2019) [11] investigated how the scour depth varies with the pier geometry and assessed the performance of the predictors of Sheppard and Renna (2010) [12], Arneson et al. (2012) [13], Moreno et al. (2015, 2016), and Amini and Mohammad (2017) [14], for the case of a complex pier with piles, a cap, and rectangular inclined columns.

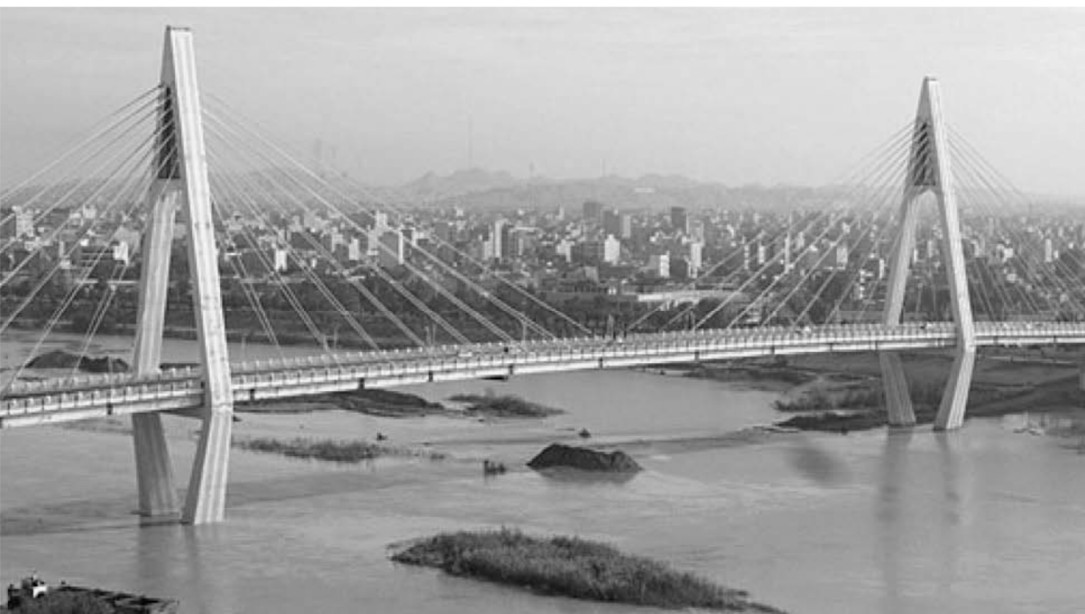

**Figure 1.** Photograph of the 8th bridge of Ahvaz on the Karun river, Iran.

Protecting bridge piers from scouring is a vital issue for the safety of bridges at the design stage [15]. Furthermore, scour countermeasures may be also installed after a bridge's construction to reduce its vulnerability during floods, thus ensuring its serviceability. Given the importance of reducing the scour depth around bridge piers, the performance of different proposed countermeasures has been assessed in various literature studies. In general, the countermeasures against pier scour are broadly classified into two categories:

(1) bed-armoring countermeasures, and (2) flow-altering countermeasures. The former are tetrapods, dolos, riprap rocks, mats, and bags, gabions and Reno mattresses (e.g., [15–20]) that increase the resistance of the riverbed to scour. Flow-altering devices are, instead, collars, slots, vanes and sacrificial piles are installed at the pier to weaken the local flow field (e.g., [19,21–23]). The selection of the most suitable scour countermeasures (or a combination of countermeasures) depends on the flow characteristics and geotechnical conditions of the riverbed at any selected site.

Among the various countermeasures that may be installed at a bridge site to mitigate local scour, such as, for example, collars (Zarrati et al., 2004) [24] or riprap (Melville et al. 2007 [18]; Cardoso and Fael 2009 [25]; Cardoso et al., 2010 [20]), some researchers have explored the possibility of using a bed sill (e.g., Chiew and Lim, 2003 [26]; Grimaldi et al., 2009a [21], 2009b [22]). The latter is commonly used to control the general scouring of a riverbed by reducing the bed slope and the flow velocity through the creation of a non-erodible section. In this way, riverbed lowering upstream of the sill is largely inhibited. Therefore, a sill is mostly used to degrade riverbeds (see an example in Figure 2). Grimaldi et al. (2009a) [21] instead investigated the efficiency of sills in reducing the local scour around a cylindrical bridge pier. According to their results, a bed sill installed at a short distance downstream of the pier reduced the maximum scour depth by around 26% in front of the pier and could reduce the scour area and volume by more than 80%. Gaudio et al. (2012) [27] experimentally investigated five combined flow-altering countermeasures against bridge pier scour including submerged vanes and a bed sill, a slot and sacrificial piles, a collar and sacrificial piles, a slot and a collar, and a bed sill and a collar; the combination of a bed sill and a collar had the best performance in reducing the maximum scour depth. Tafarojnoruz et al. (2012) [28] used a bed-sill and collar to reduce the scour depth around circular and rectangular piers under steady and unsteady flow conditions. Saadati Pacheh Kenari et al. (2014) [29] investigated the effect of sill location on the scour depth around a group of inclined piers placed on a rectangular foundation, finding that an upstream sill had the largest effect on reducing the maximum scour depth. In fact, for any top-level of foundation, the scour depth, on average, decreased by 22%, 18% and 15%, for a sill placed at the upstream edge of the pier, at the pier mid-length, and at the downstream edge of the pier, respectively. Wang et al. (2018) [30], finally, investigated the effects of a downstream submerged weir on local scour at bridge piers under clear-water and live bed conditions. For clear-water scour, a downstream submerged weir could significantly decrease the scour depth at a pier when installed close to it; for live-bed scour, furthermore, the installation of a downstream submerged weir could cause upstream bed aggradation and also a reduction in the scour depth at the pier.

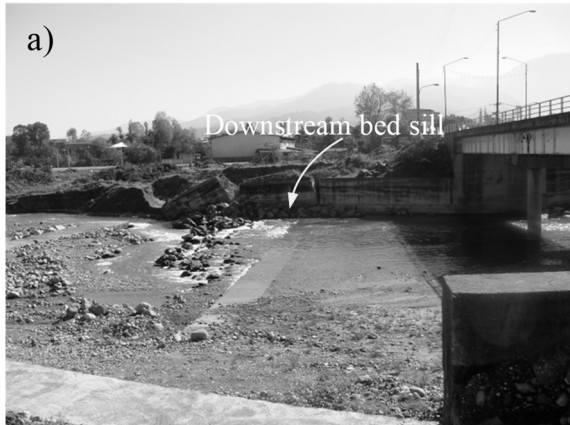 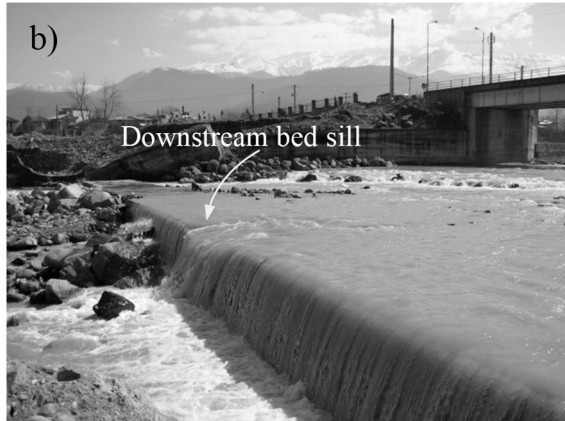

**Figure 2.** Effect of a bed sill on protecting the Machian bridge at a degrading reach of the Polroud river, Guilan province, Iran. Photographs taken in (**a**) 2010 and (**b**) 2011.

The objective of this manuscript is to test a scour countermeasure for a complex pier with inclined columns footed on a pile cap and an array of piles beneath the cap, a

geometry that has never been studied before in combination with a scour countermeasure. Furthermore, among the several countermeasures that could be used for the purpose, the manuscript explores the possibility to use a sill as a scour countermeasure, following the encouraging results documented in the literature for other geometries. An extensive (more than 140 runs) experimental campaign is presented, with runs for buried and exposed pile caps with different thickness, pile diameters and arrangements. Different sill placements (upstream, at mid-pier or downstream) are also considered. The paper, whose main focus is on how the scour reductions vary with the sill location, presents the experimental results and provides preliminary recommendations and caveats for the use of this kind of countermeasure at the investigated complex pier.

## 2. Dimensionless Framework for Analysis

The geometry of the complex pier under investigation is depicted in Figure 3. The time-dependent maximum scour depth ($d_s$) at a pier aligned with the flow is represented here by a functional relationship:

$$d_s = f_1(y, U, B, D, D_{pc}, L_{pc}, T_{pc}, Z, d_p, m, n, X_s, Z_s, D_{50}, \sigma_g, \rho_s, \rho, \mu, g, \alpha, t) \quad (1)$$

where: $y$ = flow depth; $U$ = section-averaged flow velocity; $B$ = flume width; $D$ = column width; $D_{pc}$ and $L_{pc}$ = transverse and stream-wise dimensions of the pile cap; $T_{pc}$ = thickness of the pile cap; $Z$ = elevation of the pile cap above the non-scoured bed elevation; $d_p$ = diameter of piles; $m$ = number of piles in the stream-wise direction; $n$ = number of piles in the transverse direction; $X_s$ = stream-wise coordinate of the sill, measured from the upstream edge of the pile cap; $Z_s$ = elevation of the sill above the non-scoured bed level; $D_{50}$, $\sigma_g$ and $\rho_s$ = median size, uniformity coefficient and density of sediment; $\rho$ and $\mu$ = density and viscosity of water; $g$ = acceleration due to gravity; $\alpha$ = column inclination; $t$ = time.

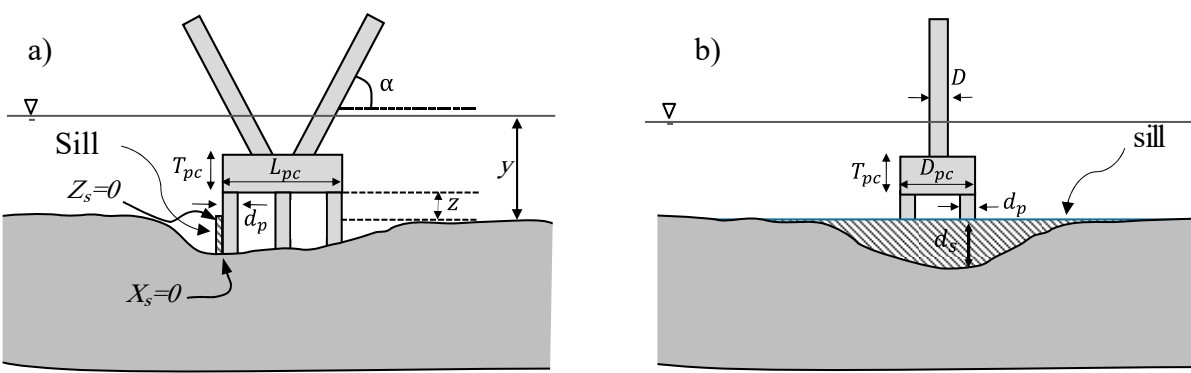

**Figure 3.** (**a**) Longitudinal and (**b**) transverse definition sketches of the investigated complex pier; sketches are for $X_s = 0$ and $Z_s = 0$.

The application of the Buckingham theorem using ($\rho, U, y$) as repeated variables yields:

$$\frac{d_s}{y} = f_2\left(\frac{B}{y}, \frac{D}{y}, \frac{D_{pc}}{y}, \frac{L_{pc}}{y}, \frac{T_{pc}}{y}, \frac{Z}{y}, \frac{d_p}{y}, m, n, \frac{X_s}{y}, \frac{Z_s}{y}, \frac{D_{50}}{y}, \sigma_g, \frac{\rho_s}{\rho}, \frac{\rho U y}{\mu}, \frac{U}{\sqrt{gy}}, \alpha, \frac{tU}{y}\right) \quad (2)$$

Then, $D$ can be used instead of $y$ as a length scale, provided that $D/y$ is present in the list of control parameters; furthermore, $L_{pc}$ can be used as a scaling factor for $X_s$:

$$\frac{d_s}{D} = f_3\left(\frac{B}{D}, \frac{D}{y}, \frac{D_{pc}}{D}, \frac{L_{pc}}{D}, \frac{T_{pc}}{D}, \frac{Z}{D}, \frac{d_p}{D}, m, n, \frac{X_s}{L_{pc}}, \frac{Z_s}{D}, \frac{D_{50}}{D}, \sigma_g, \frac{\rho_s}{\rho}, \frac{\rho U y}{\mu}, \frac{U}{\sqrt{gy}}, \alpha, \frac{tU}{D}\right) \quad (3)$$

After presenting the experimental campaign and procedure, the functional relationship will be further simplified, considering only the control parameters whose effect was accounted for in this study.

## 3. Experimental Setup

The scour experiments for the present study were carried out at the hydraulic and physical hydraulic modeling lab of the University of Guilan, Rasht, Iran, using a rectangular recirculation flume that was 8.4-m long, 0.89-m wide, and 1-m deep (Figure 4). The walls and bed of the flume were made of glass and iron panels, respectively.

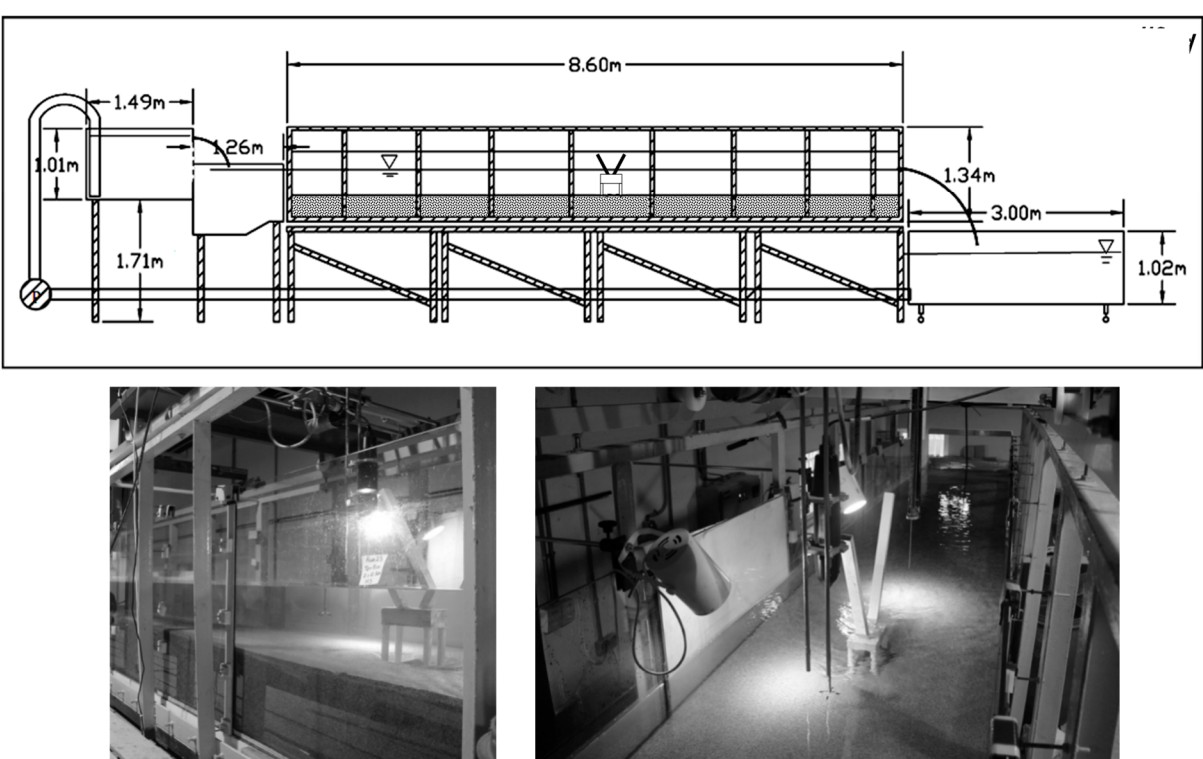

**Figure 4.** Scheme and sample photograph of the experimental flume.

The sediment used in this study was uniform sand with a particle size of 0.7 mm, large enough to avoid ripple formation. This sand was used to fill a recess section, 1.5 m long and 0.3 m deep, where the scour process would take place (the recess dimensions were enough for the development of the scour and deposition morphology downstream of the pier). The upstream and downstream reaches of the flume were covered with coarser sand with a size of 1–2 mm. The recess section and the pier model (described below) were sufficiently distant from the flume inlet to ensure that the scour process was induced by a developed flow (with the development distance computed using the formula by Kirkgöz and Ardichoğlu, 1997 [31]). Moreover, Figure 5 demonstrates good similarity between velocity profiles (the hydrodynamic condition is specified in the next section) taken at mid-flume axis for two locations, 0.5 m, and 1.5 m upstream of the pier.

A flow straightener was placed at the entrance of the flume to avoid inlet effects on the flow. To regulate the depth of water in the flume, a butterfly gate was installed at the outlet section. A centrifuge pump was used to supply a flow rate of up to 70 L/s. A motor speed controller was used to adjust the electromotor of the pump, enabling the flow discharge to be quickly and accurately adjusted. The flow rate was measured by an ultrasonic flow meter with a precision of ±0.01 L/s.

The bridge pier model consisted of two rectangular columns with section sides of 2.5 and 3.5 cm (the latter is $D$ above); these columns had an inclination of 28° to the vertical based on the prototype case, i.e., the 8th bridge of Ahvaz on the Karun river, Iran and were placed on a pile cap with a 10 cm width ($D_{pc}$), 16 cm length ($L_{pc}$), 3 or 5 cm thickness ($T_{pc}$). Below the cap, piles were present with diameters of 2 or 3 cm ($d_p$). We used two rows of piles ($n = 2$) and 2 or 3 piles in a row (thus $m = 2$ or 3). The ratios of the pier dimensions to the particle size were $d_p/D_{50} = 29$ or 43, $D/D_{50} = 50$, and $D_{pc}/D_{50} = 143$. The pier model

was complemented by a sill as a scour countermeasure. The sill model consisted of a Plexiglas sheet with a thickness of 0.06 m, width of 0.89 m (equal to the full channel width), and height of 0.3 m (equal to the depth of the recess section), which was vertically mounted upstream, middle and downstream of the pile cap, corresponding to $X_s/L_{pc}$ = 0, 0.5 and 1. The sill elevation was always equal to the elevation of the non-scoured bed (so that $Z_s$ = 0).

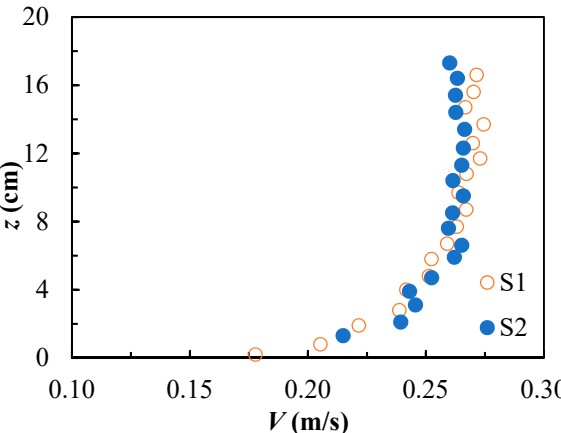

**Figure 5.** Velocity profiles taken at mid-axis, 0.5 m (S1) and 1.5 m (S2) upstream of the pier.

Constriction effects were excluded because the maximum transverse size of the pier, corresponding to $D_{pc}$, spanned 11% of the flume width. For example, Raudkivi and Ettema (1983) [32] mentioned that the effect of the sidewalls on the local scour process is negligible for flume widths larger than 6.25 times the pier size; in the present study, the ratio of flume width to pier element size was 8.8 and 25.1 for the pile cap and the column, respectively.

## 4. Experimental Procedure

The experiments presented in the following were performed with a single hydrodynamic condition of $y$ = 0.225 m and $U$ = 0.26 m/s. The Reynolds and Froude numbers were, respectively, $Re$ = 1.55 × $10^5$ (computed using four times the hydraulic radius as a length scale) and $Fr$ = 0.18 (computed using the water depth). Based on the chosen value of the water depth, the channel aspect ratio (width-to-depth ratio) was equal to 4, thus suggesting the possibility of an existing sidewall effect. Such an effect may impact the stress distribution along the transverse direction, as well as determine the difference between the mean flow velocity and the mid-channel velocity, at which point the flow actually impacts the pier. However, since the experiments in the present campaign were run with a single hydrodynamic configuration, there is no variability in the sidewall effect and the results of the runs are fully comparable. A similar argument may hold for viscous effects: given the small sediment diameter, a shear Reynolds number was around 10–20; thus, viscous effects could affect the scour levels (e.g., Oliveto and Hager, 2002 [33]; Lança et al., 2016 [34]). However, again, here we are comparing scour depth values for a single hydrodynamic condition.

The above values correspond to clear-water scour. In this condition, the present knowledge is that running experiments close to incipient sediment motion in the upstream reach leads to the largest possible scour depth. Furthermore, it is generally known that the determination of incipient motion conditions is affected by a high level of uncertainty (e.g., Dey, 2014 [35]). In this work, preliminary runs were conducted with variable flow velocity and a visual inspection of the particle motion for any velocity value. A range could be identified with velocities from 0.21 to 0.27 m/s, which induced different motion conditions from occasional single-particle movement to frequent sediment displacement throughout the entire recess section. The value of a threshold velocity $U_c$ was then set at 0.27 m/s. This value is lower than that obtained, for example, with the predictor of Melville and Coleman (2000) [1], with results equal to 0.36 m/s. However, a systematic comparison (Buffington and Montgomery, 1997 [36]) demonstrated that the threshold velocities determined by

visual inspection typically result in lower values compared to alternative methods. Based on the present estimation of $U_c$ and on the above-mentioned velocity, the $U/U_c$ ratio was kept at a value of 0.95 in all experiments. If one instead trusted the estimation of $U_c$ using Melville and Coleman's equation, a $U/U_c$ ratio of 0.72 would be obtained. In the second case, the scour depth values measured in this work would not be the largest measurable ones. However, one must note that the following results will demonstrate the effect of the sill as a scour countermeasure for a single hydro-dynamic condition, whatever value the $U/U_c$ ratio took in the tests.

Some preliminary scour experiments were performed for a duration of 72 h; the results of these experiments (not shown here) indicated that, at a dimensionless time, $tU/D = 1.5 \times 10^5$ (corresponding to 7 h) a scour depth as large as 70–80% of the equilibrium could be reached, while, at $tU/D = 5.4 \times 10^5$ (25 h), the scour depth was more than 90% of the equilibrium. The experiments that are presented, as listed in Table 1, were run as follows. Water was let into the channel at a low discharge to prevent undesired sediment motion before the run was intended to start. Then, the flow depth was increased to the prescribed value of 0.225 m using the butterfly gate. Finally, the flow discharge was increased to the target value of 0.054 m$^3$/s, continuously adjusting the butterfly gate to maintain a constant water depth. Scour depth values were frequently collected for 7 h using a point gauge whose shape enabled scour depth values to be taken below the cap if needed. A final scour value was taken on the morning of the next day (the experiments were not stopped during the night), corresponding to a duration of 25 h ($d_{sf}$ in Table 1). At this duration, the topography of the scour hole was surveyed using a laser distance sensor with an accuracy of ±1 mm (the use of this sensor did not enable scour depth values to be taken beneath the pile cap). A total of 144 runs was performed under different geometric pile cap values $T_{pc}$, the pile diameter $d_p$, the number of piles in a row $m$, the elevation of the pile cap $Z$, and the location of the sill $X_s$ (see Figure 6).

a)
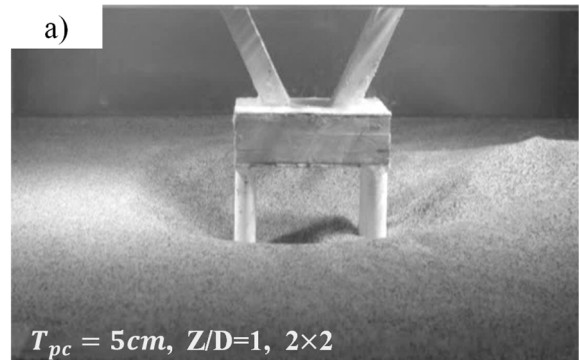
$T_{pc} = 5cm$, Z/D=1, 2×2

b)
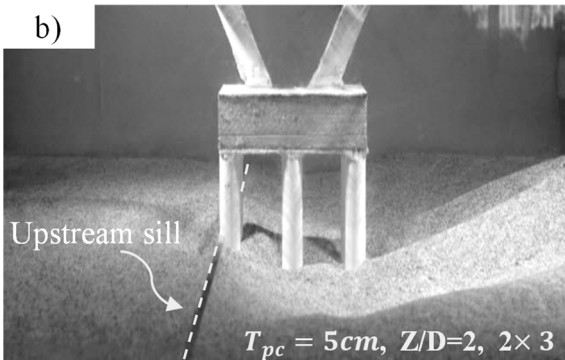
Upstream sill
$T_{pc} = 5cm$, Z/D=2, 2× 3

c)
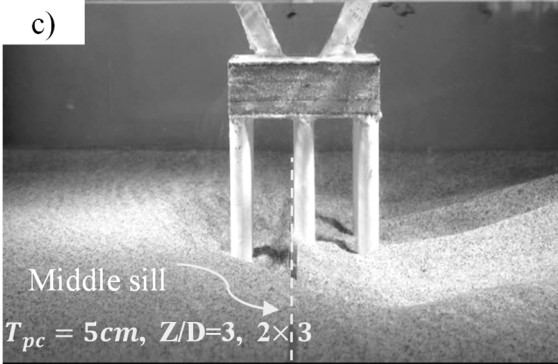
Middle sill
$T_{pc} = 5cm$, Z/D=3, 2×3

d)
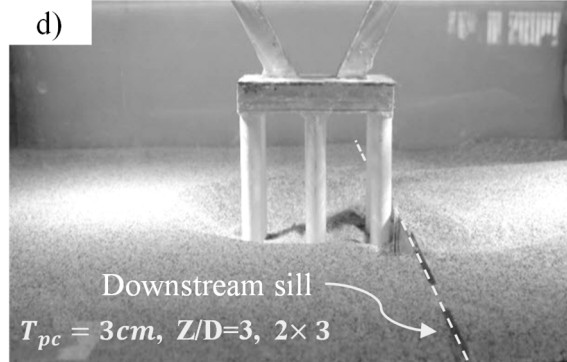
Downstream sill
$T_{pc} = 3cm$, Z/D=3, 2× 3

**Figure 6.** Sample arrangement of the pier components in different experiments: (**a**) no sill, (**b**) upstream sill ($X_s = 0$), (**c**) middle sill ($X_s = 0.5$), and (**d**) downstream sill ($X_s = 1$). The arrangements of the piles beneath the cap are furnished as $n \times m$.

**Table 1.** Details of the experimental runs; note that $d_{s5}$ is the maximum scour depth at $tU/D = 1.5 \times 10^5$.

| Run Num. | $d_p$ (m) | $m$ | $T_{pc}$ (m) | Z (m) | $X_s/L_{ps}$ | $d_{s5}$ (m) | $d_{sf}$ (m) | Run Num. | $d_p$ (m) | $m$ | $T_{pc}$ (m) | Z (m) | $X_s/L_{ps}$ | $d_{s5}$ (m) | $d_{sf}$ (m) |
|---|---|---|---|---|---|---|---|---|---|---|---|---|---|---|---|
| 1 | 0.02 | 2 | 0.03 | 0 | NS | 0.057 | 0.069 | 32 | 0.02 | 3 | 0.05 | 0.035 | 1 | 0.061 | 0.073 |
| 2 | 0.02 | 2 | 0.03 | 0 | 0 | 0.044 | 0.052 | 33 | 0.03 | 2 | 0.03 | 0.035 | NS | 0.068 | 0.083 |
| 3 | 0.02 | 2 | 0.03 | 0 | 0.5 | 0.048 | 0.06 | 34 | 0.03 | 2 | 0.03 | 0.035 | 0 | 0.043 | 0.058 |
| 4 | 0.02 | 2 | 0.03 | 0 | 1 | 0.05 | 0.062 | 35 | 0.03 | 2 | 0.03 | 0.035 | 0.5 | 0.057 | 0.068 |
| 5 | 0.02 | 2 | 0.05 | 0 | NS | 0.06 | 0.072 | 36 | 0.03 | 2 | 0.03 | 0.035 | 1 | 0.064 | 0.077 |
| 6 | 0.02 | 2 | 0.05 | 0 | 0 | 0.043 | 0.054 | 37 | 0.03 | 3 | 0.03 | 0.035 | NS | 0.07 | 0.087 |
| 7 | 0.02 | 2 | 0.05 | 0 | 0.5 | 0.053 | 0.063 | 38 | 0.03 | 3 | 0.03 | 0.035 | 0 | 0.045 | 0.059 |
| 8 | 0.02 | 2 | 0.05 | 0 | 1 | 0.055 | 0.066 | 39 | 0.03 | 3 | 0.03 | 0.035 | 0.5 | 0.059 | 0.068 |
| 9 | 0.03 | 2 | 0.03 | 0 | NS | 0.063 | 0.077 | 40 | 0.03 | 3 | 0.03 | 0.035 | 1 | 0.067 | 0.079 |
| 10 | 0.03 | 2 | 0.03 | 0 | 0 | 0.044 | 0.057 | 41 | 0.03 | 2 | 0.05 | 0.035 | NS | 0.068 | 0.08 |
| 11 | 0.03 | 2 | 0.03 | 0 | 0.5 | 0.055 | 0.066 | 42 | 0.03 | 2 | 0.05 | 0.035 | 0 | 0.043 | 0.057 |
| 12 | 0.03 | 2 | 0.03 | 0 | 1 | 0.061 | 0.07 | 43 | 0.03 | 2 | 0.05 | 0.035 | 0.5 | 0.059 | 0.069 |
| 13 | 0.03 | 2 | 0.05 | 0 | NS | 0.064 | 0.078 | 44 | 0.03 | 2 | 0.05 | 0.035 | 1 | 0.062 | 0.076 |
| 14 | 0.03 | 2 | 0.05 | 0 | 0 | 0.049 | 0.057 | 45 | 0.03 | 3 | 0.05 | 0.035 | NS | 0.07 | 0.083 |
| 15 | 0.03 | 2 | 0.05 | 0 | 0.5 | 0.055 | 0.062 | 46 | 0.03 | 3 | 0.05 | 0.035 | 0 | 0.049 | 0.06 |
| 16 | 0.03 | 2 | 0.05 | 0 | 1 | 0.057 | 0.066 | 47 | 0.03 | 3 | 0.05 | 0.035 | 0.5 | 0.06 | 0.07 |
| 17 | 0.02 | 2 | 0.03 | 0.035 | NS | 0.059 | 0.071 | 48 | 0.03 | 3 | 0.05 | 0.035 | 1 | 0.066 | 0.079 |
| 18 | 0.02 | 2 | 0.03 | 0.035 | 0 | 0.045 | 0.053 | 49 | 0.02 | 2 | 0.03 | 0.07 | NS | 0.054 | 0.065 |
| 19 | 0.02 | 2 | 0.03 | 0.035 | 0.5 | 0.052 | 0.06 | 50 | 0.02 | 2 | 0.03 | 0.07 | 0 | 0.033 | 0.043 |
| 20 | 0.02 | 2 | 0.03 | 0.035 | 1 | 0.055 | 0.062 | 51 | 0.02 | 2 | 0.03 | 0.07 | 0.5 | 0.045 | 0.056 |
| 21 | 0.02 | 3 | 0.03 | 0.035 | NS | 0.063 | 0.076 | 52 | 0.02 | 2 | 0.03 | 0.07 | 1 | 0.051 | 0.059 |
| 22 | 0.02 | 3 | 0.03 | 0.035 | 0 | 0.048 | 0.055 | 53 | 0.02 | 3 | 0.03 | 0.07 | NS | 0.06 | 0.07 |
| 23 | 0.02 | 3 | 0.03 | 0.035 | 0.5 | 0.056 | 0.067 | 54 | 0.02 | 3 | 0.03 | 0.07 | 0 | 0.038 | 0.046 |
| 24 | 0.02 | 3 | 0.03 | 0.035 | 1 | 0.06 | 0.071 | 55 | 0.02 | 3 | 0.03 | 0.07 | 0.5 | 0.048 | 0.055 |
| 25 | 0.02 | 2 | 0.05 | 0.035 | NS | 0.063 | 0.075 | 56 | 0.02 | 3 | 0.03 | 0.07 | 1 | 0.051 | 0.06 |
| 26 | 0.02 | 2 | 0.05 | 0.035 | 0 | 0.043 | 0.054 | 57 | 0.02 | 2 | 0.05 | 0.07 | NS | 0.059 | 0.07 |
| 27 | 0.02 | 2 | 0.05 | 0.035 | 0.5 | 0.054 | 0.065 | 58 | 0.02 | 2 | 0.05 | 0.07 | 0 | 0.039 | 0.048 |
| 28 | 0.02 | 2 | 0.05 | 0.035 | 1 | 0.06 | 0.072 | 59 | 0.02 | 2 | 0.05 | 0.07 | 0.5 | 0.051 | 0.058 |
| 29 | 0.02 | 3 | 0.05 | 0.035 | NS | 0.067 | 0.079 | 60 | 0.02 | 2 | 0.05 | 0.07 | 1 | 0.056 | 0.065 |
| 30 | 0.02 | 3 | 0.05 | 0.035 | 0 | 0.048 | 0.061 | 61 | 0.02 | 3 | 0.05 | 0.07 | NS | 0.065 | 0.074 |
| 31 | 0.02 | 3 | 0.05 | 0.035 | 0.5 | 0.056 | 0.068 | 62 | 0.02 | 3 | 0.05 | 0.07 | 0 | 0.044 | 0.053 |
| 63 | 0.02 | 3 | 0.05 | 0.07 | 0.5 | 0.053 | 0.062 | 96 | 0.02 | 3 | 0.05 | 0.105 | 1 | 0.06 | 0.068 |
| 64 | 0.02 | 3 | 0.05 | 0.07 | 1 | 0.059 | 0.07 | 97 | 0.03 | 2 | 0.03 | 0.105 | NS | 0.067 | 0.08 |
| 65 | 0.03 | 2 | 0.03 | 0.07 | NS | 0.068 | 0.081 | 98 | 0.03 | 2 | 0.03 | 0.105 | 0 | 0.04 | 0.054 |
| 66 | 0.03 | 2 | 0.03 | 0.07 | 0 | 0.042 | 0.056 | 99 | 0.03 | 2 | 0.03 | 0.105 | 0.5 | 0.053 | 0.063 |
| 67 | 0.03 | 2 | 0.03 | 0.07 | 0.5 | 0.056 | 0.066 | 100 | 0.03 | 2 | 0.03 | 0.105 | 1 | 0.061 | 0.07 |
| 68 | 0.03 | 2 | 0.03 | 0.07 | 1 | 0.061 | 0.072 | 101 | 0.03 | 3 | 0.03 | 0.105 | NS | 0.07 | 0.083 |
| 69 | 0.03 | 3 | 0.03 | 0.07 | NS | 0.071 | 0.085 | 102 | 0.03 | 3 | 0.03 | 0.105 | 0 | 0.043 | 0.056 |
| 70 | 0.03 | 3 | 0.03 | 0.07 | 0 | 0.047 | 0.058 | 103 | 0.03 | 3 | 0.03 | 0.105 | 0.5 | 0.057 | 0.064 |
| 71 | 0.03 | 3 | 0.03 | 0.07 | 0.5 | 0.06 | 0.067 | 104 | 0.03 | 3 | 0.03 | 0.105 | 1 | 0.064 | 0.076 |
| 72 | 0.03 | 3 | 0.03 | 0.07 | 1 | 0.065 | 0.077 | 105 | 0.03 | 2 | 0.05 | 0.105 | NS | 0.071 | 0.085 |
| 73 | 0.03 | 2 | 0.05 | 0.07 | NS | 0.069 | 0.083 | 106 | 0.03 | 2 | 0.05 | 0.105 | 0 | 0.042 | 0.054 |
| 74 | 0.03 | 2 | 0.05 | 0.07 | 0 | 0.047 | 0.053 | 107 | 0.03 | 2 | 0.05 | 0.105 | 0.5 | 0.057 | 0.065 |
| 75 | 0.03 | 2 | 0.05 | 0.07 | 0.5 | 0.057 | 0.063 | 108 | 0.03 | 2 | 0.05 | 0.105 | 1 | 0.066 | 0.072 |
| 76 | 0.03 | 2 | 0.05 | 0.07 | 1 | 0.063 | 0.071 | 109 | 0.03 | 3 | 0.05 | 0.105 | NS | 0.074 | 0.09 |
| 77 | 0.03 | 3 | 0.05 | 0.07 | NS | 0.071 | 0.087 | 110 | 0.03 | 3 | 0.05 | 0.105 | 0 | 0.05 | 0.059 |
| 78 | 0.03 | 3 | 0.05 | 0.07 | 0 | 0.052 | 0.058 | 111 | 0.03 | 3 | 0.05 | 0.105 | 0.5 | 0.062 | 0.069 |
| 79 | 0.03 | 3 | 0.05 | 0.07 | 0.5 | 0.06 | 0.69 | 112 | 0.03 | 3 | 0.05 | 0.105 | 1 | 0.068 | 0.075 |
| 80 | 0.03 | 3 | 0.05 | 0.07 | 1 | 0.065 | 0.074 | 113 | 0.02 | 2 | 0.03 | 0.14 | NS | 0.051 | 0.059 |
| 81 | 0.02 | 2 | 0.03 | 0.105 | NS | 0.054 | 0.063 | 114 | 0.02 | 2 | 0.03 | 0.14 | 0 | 0.031 | 0.037 |
| 82 | 0.02 | 2 | 0.03 | 0.105 | 0 | 0.036 | 0.04 | 115 | 0.02 | 2 | 0.03 | 0.14 | 0.5 | 0.044 | 0.05 |
| 83 | 0.02 | 2 | 0.03 | 0.105 | 0.5 | 0.047 | 0.052 | 116 | 0.02 | 2 | 0.03 | 0.14 | 1 | 0.047 | 0.052 |
| 84 | 0.02 | 2 | 0.03 | 0.105 | 1 | 0.052 | 0.057 | 117 | 0.02 | 3 | 0.03 | 0.14 | NS | 0.053 | 0.062 |
| 85 | 0.02 | 3 | 0.03 | 0.105 | NS | 0.058 | 0.067 | 118 | 0.02 | 3 | 0.03 | 0.14 | 0 | 0.032 | 0.039 |
| 86 | 0.02 | 3 | 0.03 | 0.105 | 0 | 0.037 | 0.043 | 119 | 0.02 | 3 | 0.03 | 0.14 | 0.5 | 0.045 | 0.05 |
| 87 | 0.02 | 3 | 0.03 | 0.105 | 0.5 | 0.048 | 0.053 | 120 | 0.02 | 3 | 0.03 | 0.14 | 1 | 0.05 | 0.056 |
| 88 | 0.02 | 3 | 0.03 | 0.105 | 1 | 0.052 | 0.058 | 121 | 0.02 | 2 | 0.05 | 0.14 | NS | 0.057 | 0.064 |

**Table 1.** *Cont.*

| Run Num. | $d_p$ (m) | $m$ | $T_{pc}$ (m) | $Z$ (m) | $X_s/L_{ps}$ | $d_{s5}$ (m) | $d_{sf}$ (m) | Run Num. | $d_p$ (m) | $m$ | $T_{pc}$ (m) | $Z$ (m) | $X_s/L_{ps}$ | $d_{s5}$ (m) | $d_{sf}$ (m) |
|---|---|---|---|---|---|---|---|---|---|---|---|---|---|---|---|
| 89 | 0.02 | 2 | 0.05 | 0.105 | NS | 0.057 | 0.065 | 122 | 0.02 | 2 | 0.05 | 0.14 | 0 | 0.032 | 0.04 |
| 90 | 0.02 | 2 | 0.05 | 0.105 | 0 | 0.03 | 0.043 | 123 | 0.02 | 2 | 0.05 | 0.14 | 0.5 | 0.047 | 0.051 |
| 91 | 0.02 | 2 | 0.05 | 0.105 | 0.5 | 0.047 | 0.053 | 124 | 0.02 | 2 | 0.05 | 0.14 | 1 | 0.051 | 0.055 |
| 92 | 0.02 | 2 | 0.05 | 0.105 | 1 | 0.053 | 0.062 | 125 | 0.02 | 3 | 0.05 | 0.14 | NS | 0.06 | 0.069 |
| 93 | 0.02 | 3 | 0.05 | 0.105 | NS | 0.062 | 0.071 | 126 | 0.02 | 3 | 0.05 | 0.14 | 0 | 0.036 | 0.043 |
| 94 | 0.02 | 3 | 0.05 | 0.105 | 0 | 0.034 | 0.046 | 127 | 0.02 | 3 | 0.05 | 0.14 | 0.5 | 0.049 | 0.053 |
| 95 | 0.02 | 3 | 0.05 | 0.105 | 0.5 | 0.051 | 0.059 | 128 | 0.02 | 3 | 0.05 | 0.14 | 1 | 0.053 | 0.06 |
| 129 | 0.03 | 2 | 0.03 | 0.14 | NS | 0.065 | 0.078 | 137 | 0.03 | 2 | 0.05 | 0.14 | NS | 0.072 | 0.089 |
| 130 | 0.03 | 2 | 0.03 | 0.14 | 0 | 0.041 | 0.051 | 138 | 0.03 | 2 | 0.05 | 0.14 | 0 | 0.044 | 0.054 |
| 131 | 0.03 | 2 | 0.03 | 0.14 | 0.5 | 0.054 | 0.06 | 139 | 0.03 | 2 | 0.05 | 0.14 | 0.5 | 0.06 | 0.069 |
| 132 | 0.03 | 2 | 0.03 | 0.14 | 1 | 0.06 | 0.069 | 140 | 0.03 | 2 | 0.05 | 0.14 | 1 | 0.066 | 0.073 |
| 133 | 0.03 | 3 | 0.03 | 0.14 | NS | 0.068 | 0.08 | 141 | 0.03 | 3 | 0.05 | 0.14 | NS | 0.076 | 0.093 |
| 134 | 0.03 | 3 | 0.03 | 0.14 | 0 | 0.043 | 0.053 | 142 | 0.03 | 3 | 0.05 | 0.14 | 0 | 0.051 | 0.06 |
| 135 | 0.03 | 3 | 0.03 | 0.14 | 0.5 | 0.058 | 0.062 | 143 | 0.03 | 3 | 0.05 | 0.14 | 0.5 | 0.065 | 0.072 |
| 136 | 0.03 | 3 | 0.03 | 0.14 | 1 | 0.062 | 0.074 | 144 | 0.03 | 3 | 0.05 | 0.14 | 1 | 0.07 | 0.077 |

## 5. Results

### 5.1. Phenomenological Description and Temporal Evolution of the Maximum Scour Depth

The key features of the scour process resembled the typical descriptions in the literature, with the down-flow, and resulting vortices. Some changes in the phenomenology could be observed for the varying elevations of the pile cap. A pile cap at $Z = 0$ acted as a large obstacle at the sediment bed but also intercepted the down-flow from the columns above. For $Z > 0$, the flow could pass under the pile cap and horseshoe vortices appeared around each pile. By further increasing $Z/D$ to 2–4, the amount of flow that could pass through the pile array was progressively increased, and the vortices around the piles grew stronger.

Examples of the temporal evolution of the maximum (within the scour hole) dimensionless scour depth for the performed runs are shown in Figure 7. The panels in the figure correspond to different values of the cap elevation $Z/D$, selecting runs with $d_p = 0.03$ m, $m = 2$, and $T_{pc} = 0.05$ m; in each panel, the run succession is from an unprotected pier to $X_s/L_{pc} = 0$, 0.5, and then 1. All the temporal trends present the usual decrease in the scour rate and a final tendency towards equilibrium (note that the values depicted in this Figure are for the first 7 h of run). The temporal curves for the unprotected piers (shown with dashed lines) were not much different to, and sometimes overlapped, those with a sill placed at the downstream end of the cap (last curve in each panel), indicating that the sill was not effective in reducing the scour depth in those cases. By contrast, the sill was an effective scour countermeasure when placed at the two other locations. One can furnish quantitative indicators of the countermeasure performance, considering some reference instants; in this paragraph, values are given for two-dimensional times $tU/D$ of $3 \times 10^4$ and $10^5$ (we leave an analysis of the long-term scour values to the following subsection). Data inspection (considering all the runs, not only those in the diagrams) revealed that, for $Z/D = 0$, the value of the maximum scour depth at the two considered dimensionless times could be reduced by 6–30% (at $tU/D = 3 \times 10^4$) and 10–25% (at $tU/D = 10^5$). The results for other cap elevations were similar: the reduction in the scour depth for different sill positions was in the range of 3–47% and 4–40% for $Z/D = 1$, 4–43% and 7–40% for $Z/D = 2$, 3–48% and 5–49% for $Z/D = 3$, and 2–50% and 6–45% for $Z/D = 4$. The scour percentages for the two times are similar, with some decrease from the former to the latter due to the temporal increase in the scour depth. The analysis for the final scour depth values in each test is presented in the following subsection.

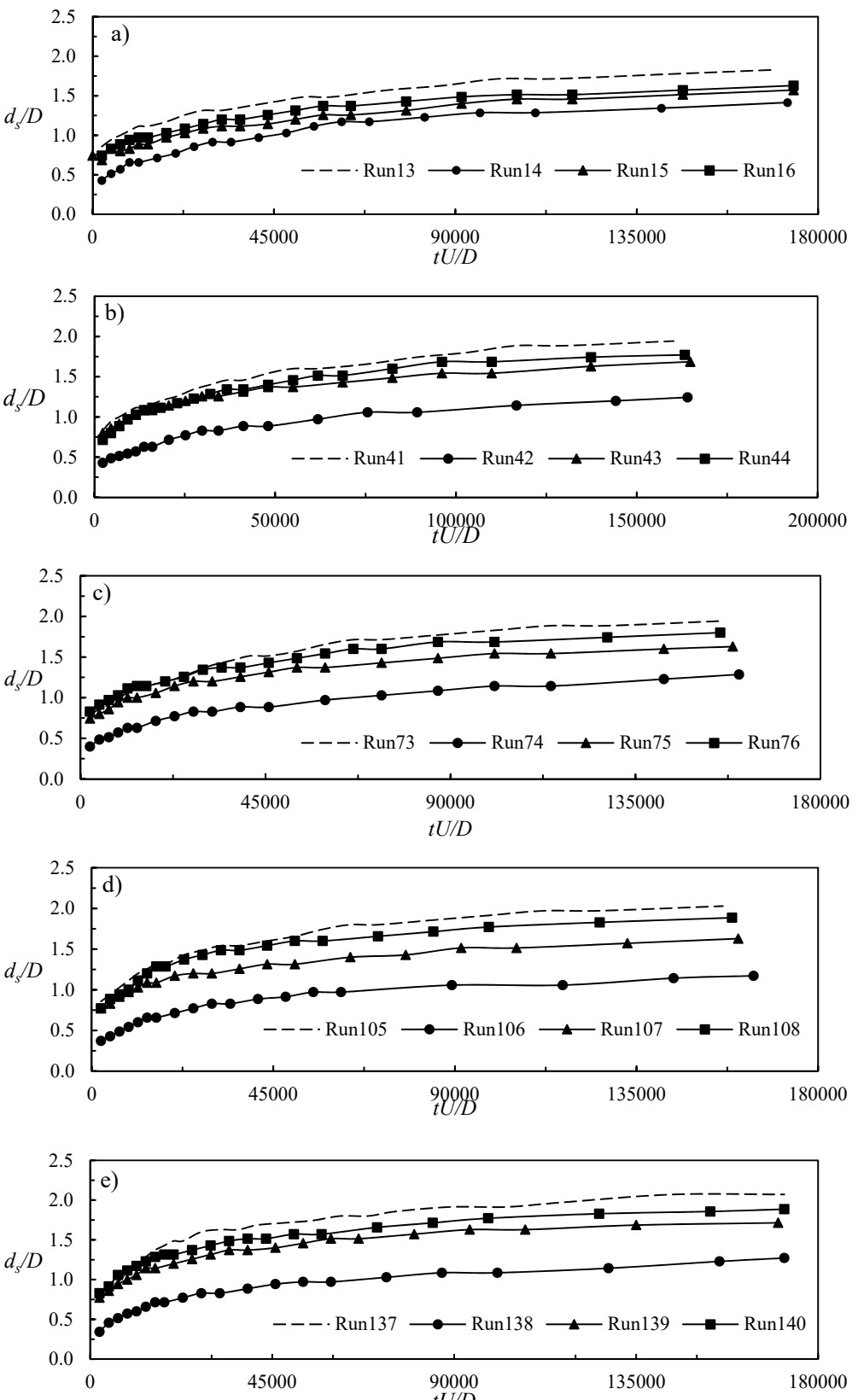

**Figure 7.** Temporal development of the maximum scour depth for (**a**) $Z/D = 0$, (**b**) $Z/D = 1$, (**c**) $Z/D = 2$, (**d**) $Z/D = 3$, and (**e**) $Z/D = 4$. In each plot, the first run is for no sill and the others are ordered as $X_s/L_{pc} = 0$, 0.5, and 1.

Effect of a Sill Location on the Final Scour Depth

Many parameters of (3) were kept constant in the experimentation. Therefore, the data analysis of the present sub-section was based on a simplified version of (3), including only parameters that were varied in this study:

$$\frac{d_s}{D} = f_3\left(\frac{T_{pc}}{D}, \frac{Z}{D}, \frac{d_p}{D}, m, \frac{X_s}{L_{pc}}, \frac{tU}{D}\right) \qquad (4)$$

In a simple application of the Buckingham theorem, we chose to maintain the column width $D$ as a unique length scale. One could object, as the column is the thinnest element of the complex pier and, moreover, the farthest from the bed; therefore, other components could provide dominant length scales for the scour process. While this is reasonable, we remark at this point that the major aim of the manuscript is to find an optimal value for $X_s$, and any length of scaling would similarly reveal this.

The effect of a sill location ($X_s/L_{pc} = 0$, 0.5 and 1, corresponding to an upstream, middle and downstream sill, respectively) on the maximum scour depth at end of tests, $d_{sf}$, is shown in Figures 8 and 9 for narrow ($d_p/D = 0.57$) and thick ($d_p/D = 0.85$) foundation piles, respectively. Various combinations of cap thickness, pile number and cap elevation are considered. It should be noted that, for $Z/D = 0$, only the pile arrangement of $2 \times 2$ was considered, because the scour depth could not extend beneath the pile cap; therefore, cap arrangement had no effect on the scour depth. All the plots showed a consistent trend, demonstrating that the highest sill performance as a scour countermeasure was obtained for $X_s/L_{pc} = 0$. On average, we obtained mean reductions in the maximum scour depth of a bit more than 30% for $X_s/L_{pc} = 0$, around 20% for $X_s/L_{pc} = 0.5$, and of around 10% for $X_s/L_{pc} = 1$. This is the most important finding of the analysis, and will be discussed and compared to the existing literature in a later section.

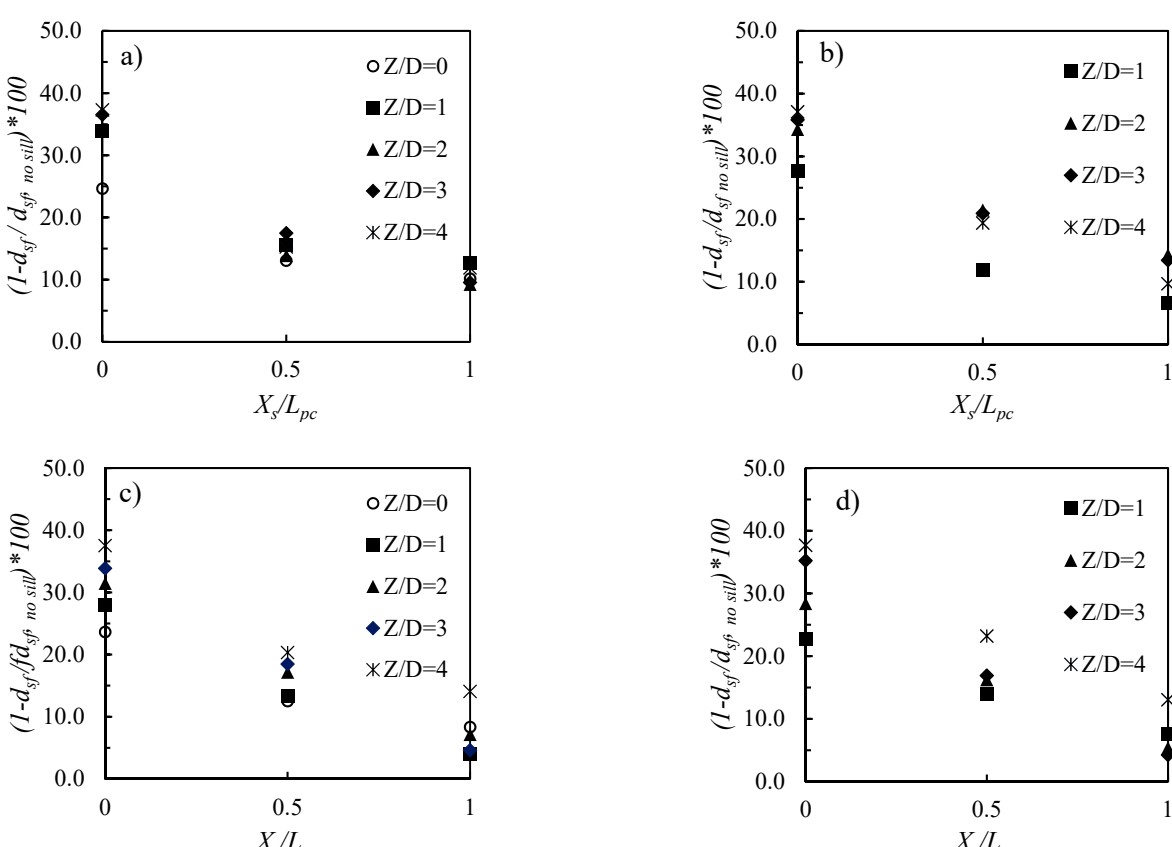

**Figure 8.** Comparison of maximum scour depth in conditions with and without sill for $T_{pc}/D = 0.85$, $d_p/D = 0.57$, (**a**) $m = 2$, (**b**) $m = 3$ and $T_{pc}/D = 1.42$, $d_p/D = 0.57$, (**c**) $m = 2$, (**d**) $m = 3$.

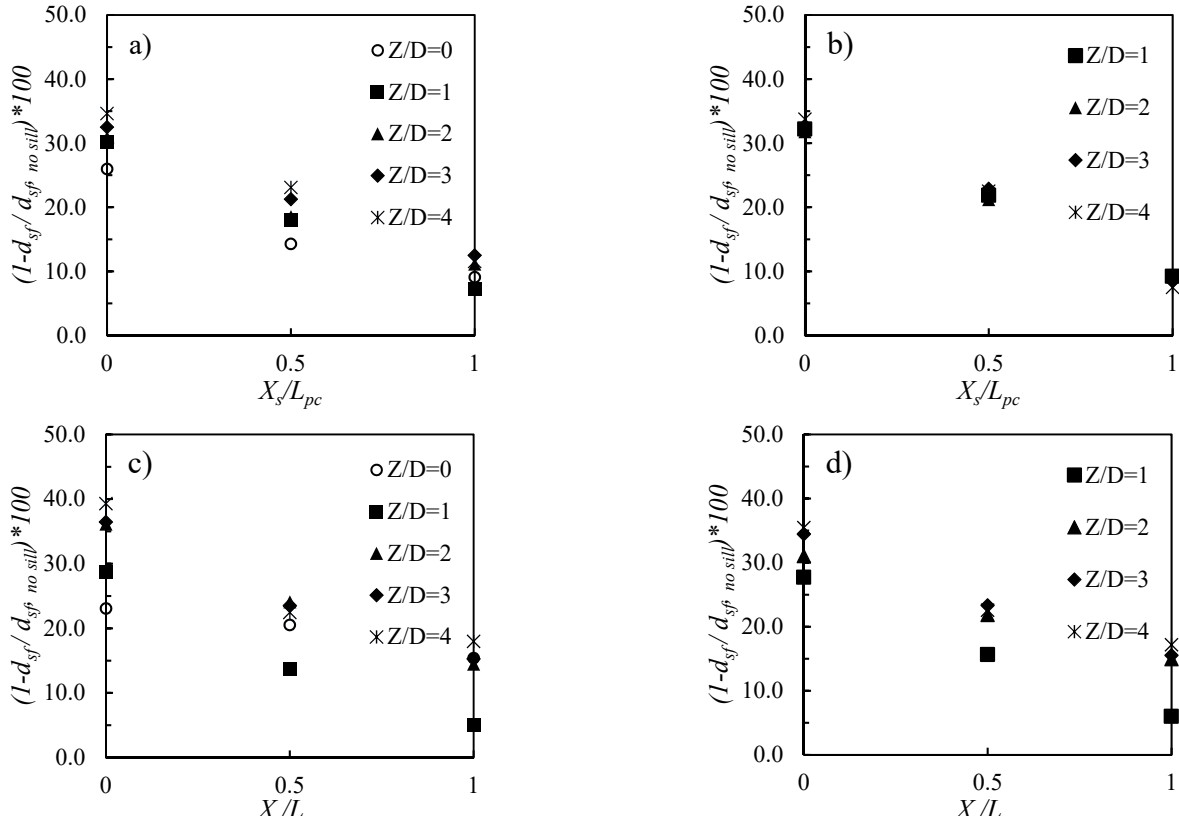

**Figure 9.** Comparison of maximum scour depth in conditions with and without sill for $T_{pc}/D = 0.85$, $d_p/D = 0.85$, (**a**) $m = 2$, (**b**) $m = 3$ and $T_{pc}/D = 1.42$, $d_p/D = 0.85$, (**c**) $m = 2$, (**d**) $m = 3$.

According to the performance of sills regarding the reduction in scour depth, the upstream sill had the greatest effect on the reduction in scour depth. For $Z/D = 0$. The change in pile cap thickness and pile diameter had a similar effect on the sills' performance and the effect of a change in pile cap thickness in the relative pile diameter of 0.85 was more noticeable for the middle and downstream sills. The effect of changing the relative diameter of the pile was also more noticeable for the middle and downstream sills in the relative pile cap thickness of 1.42. For $Z/D = 1$, the change in the pile cap thickness had the greatest effect on sills performance. In general, increasing the pile cap thickness caused a reduction in the sills 'performance. Increasing the pile diameter resulted in the increase in the sills' performance, except in the case of $T_{pc}/D = 0.85$ and $N = 2$, in which the performance of the upstream and downstream sills decreased with increasing pile diameter. The increase in the number of piles, from 2 to 3, had the lowest effect on sills' efficiency.

For $Z/D = 2$, a change in pile diameter had the strongest effect on sills' performance and the increase in the number of piles from 2 to 3 had the lowest effect on sills' efficiency. At this level, increasing the pile diameter caused an increase in the sills' performance, except for $T_{pc}/D = 0.85$ and $N = 3$, in which increasing the pile diameter caused a reduction in the sills' performance. By increasing the pile number, the sills' performance fluctuated. For $d_p/D = 0.85$, by increasing the pile cap thickness, sills' performance increased, and $T_{pc}/D = 0.85$ and $N = 3$ decreased. For $T_{pc}/D = 0.85$ and $N = 2$, the performance of the upstream and downstream sills decreased with increases in the pile diameter. For $Z/D = 3$, a change in pile cap thickness had the strongest effect on sills' performance, and an increase in the number of piles from 2 to 3 had the lowest effect on sills' efficiency. A comparison of the results showed that increasing the pile diameter led to a decrease in the upstream sills' performance, except for $T_{pc}/D = 1.42$ and $N = 2$. At this level, for $d_{pc}/D = 0.85$, increasing the pile cap thickness caused a decrease in the sills' performance and, for $d_{pc}/D = 1.42$, an increase of the pile cap thickness led to an increase in the sills' performance.

For $Z/D$ = 4, a change in pile cap thickness had the strongest effect on the sills' performance, and an increase in the number of piles from 2 to 3 had the lowest effect on sills' efficiency. In this level, increasing the pile cap thickness caused an increase in the sills' performance for all cases and by increasing the pile diameter for $T_{pc}/D$ = 1.42, the sill performance increased. At this level, the effect of increasing the pile number on the sills' performance was negligible.

### 5.2. Dimensions of the Scour Hole

A study of the dimensions of the scour hole was carried out to complement the analysis of the maximum depth. Figure 10 presents sample scour topographies, as surveyed after the end of four runs (all for $Z/D$ = 1, $T_{pc}/D$ = 1.42, $d_p/D$ = 0.85, and $m$ = 3). The contour maps show that upstream or middle sills increased the width of the scour hole compared to the run with an unprotected pier, while the use of a downstream sill determined a similar hole width to that in the case without a countermeasure.

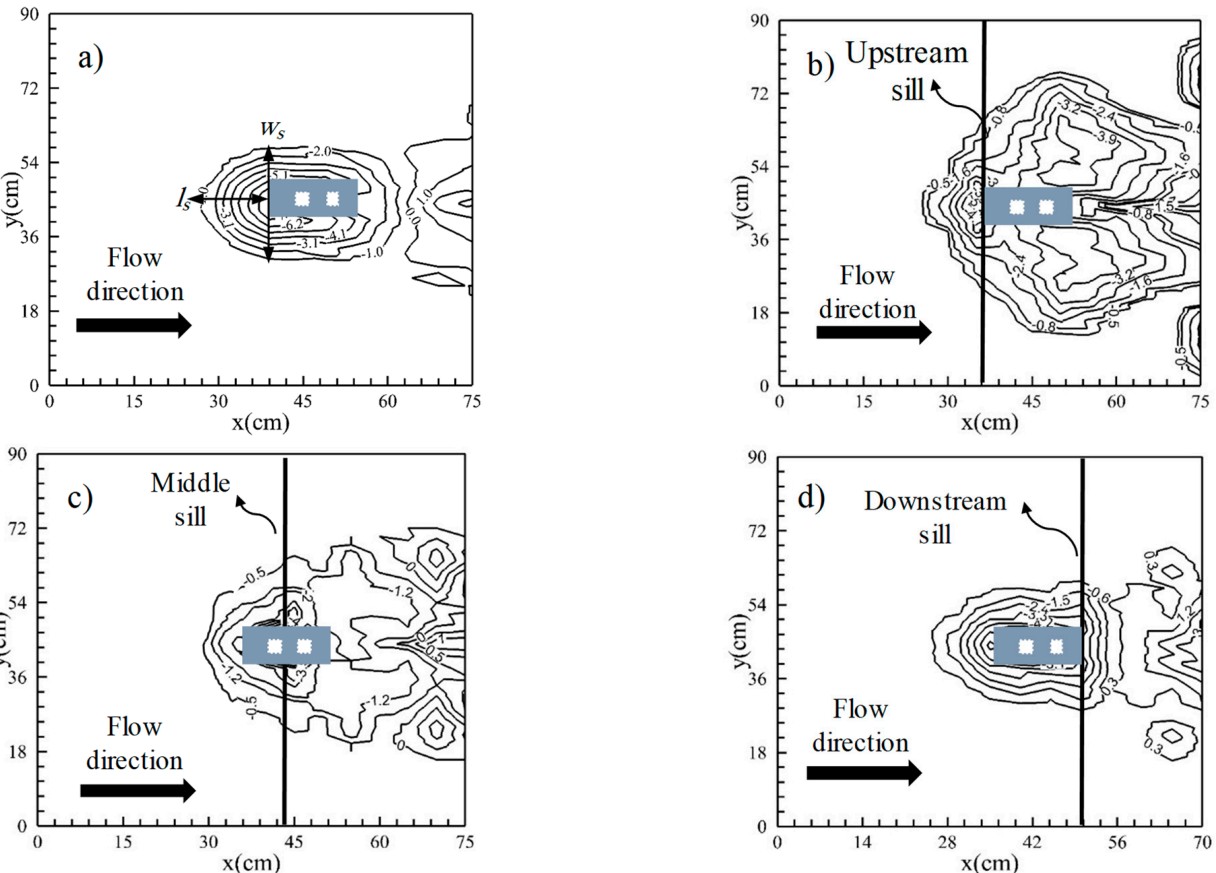

**Figure 10.** The topography of the final scour bed for $Z/D$ = 1, $T_{pc}/D$ = 1.42, $d_p/D$ = 0.85 and pile arrangement of 2 × 3, (**a**) no sill, (**b**) upstream sill, (**c**) middle sill and (**d**) downstream sill. Note that the units of contours are cm.

The results obtained from all the runs are listed in Table 2 in terms of relative width ($a_w = w_s/d_{sf}$) and length ($a_l = l_s/d_{sf}$) of the scour hole. Figure 10 clarifies how these dimensions were measured: $w_s$ is the maximum width of the scour hole, while $l_s$ is the stream-wise distance from the upstream edge of the hole to the location of maximum scour. Table 2 contains 40 values, from a lower number of experiments than the total number of 144 runs; this is because, for any combination of $Z/D$, $d_p/D$, $T_{pc}/D$ and $m$, the Table includes the maximum and minimum values of the relative width and length of the scour hole. Figure 11 presents, as Figures 8 and 9 did for the scour depth, the effect of a sill location on the hole dimensions; given how the data of Table 2 were selected, the plots

identify the widest range of variability in the shape parameters. In the plots, there is no distinction between the runs with different control parameters ($d_p/D$, $T_{pc}/D$, $m$, $Z/D$), considering that the dependency on these parameters is weaker than that on $X_s/L_{pc}$ (as it was for the scour depth). From the plots, one can see that the hole length was not as variable, although there was some decrease when the sill was moved from upstream to downstream. The hole width, instead, varied significantly with the sill location and the plot confirms the latter to be the most influential parameter. It is finally noted that the values obtained for $X_s/L_{pc} = 1$ (downstream sill) were, for both the width and length of the scour hole, similar to those obtained for the unprotected pier.

**Table 2.** Maximum and minimum values of the scour hole dimensions.

| Relative Scour Hole Width ($a_w$) | $T_{pc}/D$ | $d_p/D$ | $m$ | Relative Scour Hole Length ($a_l$) | $T_{pc}/D$ | $d_p/D$ | $m$ | Sill Position ($X_s/L_{pc}$) | $Z/D$ |
|---|---|---|---|---|---|---|---|---|---|
| 3.06 | 1.42 | 0.57 | 2 | 1.88 | 1.42 | 0.57 | 2 | No sill | |
| 2.82 | 1.42 | 0.85 | 2 | 1.79 | 1.42 | 0.85 | 2 | No sill | |
| 5.06 | 0.85 | 0.85 | 2 | 1.92 | 1.42 | 0.85 | 2 | 0 | |
| 5.38 | 0.85 | 0.57 | 2 | 2.11 | 0.85 | 0.85 | 2 | 0 | 0 |
| 4.44 | 1.42 | 0.57 | 2 | 1.82 | 0.85 | 0.85 | 2 | 0.5 | |
| 4.84 | 1.42 | 0.85 | 2 | 1.94 | 1.42 | 0.85 | 2 | 0.5 | |
| 3.23 | 0.85 | 0.57 | 2 | 1.77 | 0.85 | 0.57 | 2 | 1 | |
| 2.86 | 0.85 | 0.85 | 2 | 1.89 | 1.42 | 0.85 | 2 | 1 | |
| 2.65 | 0.85 | 0.85 | 2 | 1.63 | 1.42 | 0.85 | 2 | No sill | |
| 3.04 | 1.42 | 0.57 | 3 | 1.9 | 1.42 | 0.57 | 3 | No sill | |
| 5.95 | 0.85 | 0.57 | 2 | 2.13 | 0.85 | 0.57 | 2 | 0 | |
| 4.92 | 1.42 | 0.57 | 3 | 1.92 | 1.42 | 0.85 | 3 | 0 | 1 |
| 4.1 | 1.42 | 0.57 | 3 | 1.76 | 0.85 | 0.85 | 2 | 0.5 | |
| 4.7 | 0.85 | 0.85 | 3 | 1.91 | 0.85 | 0.85 | 3 | 0.5 | |
| 2.53 | 1.42 | 0.85 | 3 | 1.65 | 1.42 | 0.85 | 3 | 1 | |
| 3.30 | 0.85 | 0.57 | 2 | 1.85 | 0.85 | 0.57 | 2 | 1 | |
| 2.53 | 1.42 | 0.85 | 3 | 1.73 | 0.85 | 0.85 | 2 | No sill | |
| 3.15 | 1.42 | 0.57 | 2 | 1.92 | 0.85 | 0.57 | 2 | No sill | |
| 5.36 | 0.85 | 0.85 | 2 | 1.89 | 1.42 | 0.85 | 2 | 0 | |
| 6.97 | 0.85 | 0.57 | 2 | 2.11 | 1.42 | 0.85 | 3 | 0 | 2 |
| 4.44 | 1.42 | 0.85 | 2 | 1.79 | 0.85 | 0.57 | 2 | 0.5 | |
| 4.82 | 1.42 | 0.57 | 2 | 2.02 | 1.42 | 0.57 | 3 | 0.5 | |
| 2.82 | 1.42 | 0.85 | 2 | 1.69 | 0.85 | 0.57 | 2 | 1 | |
| 3.67 | 0.85 | 0.57 | 3 | 1.97 | 1.42 | 0.85 | 2 | 1 | |
| 2.5 | 0.85 | 0.85 | 2 | 1.63 | 0.85 | 0.85 | 2 | No sill | |
| 3.18 | 0.85 | 0.57 | 2 | 1.9 | 0.85 | 0.57 | 2 | No sill | |
| 5.36 | 0.85 | 0.85 | 3 | 1.86 | 1.42 | 0.57 | 2 | 0 | |
| 6.5 | 0.85 | 0.57 | 3 | 2.12 | 1.42 | 0.85 | 3 | 0 | 3 |
| 4.64 | 1.42 | 0.85 | 3 | 1.7 | 1.42 | 0.57 | 2 | 0.5 | |
| 5.28 | 0.85 | 0.57 | 3 | 2 | 1.42 | 0.85 | 2 | 0.5 | |
| 2.63 | 0.85 | 0.85 | 3 | 1.61 | 1.42 | 0.57 | 2 | 1 | |
| 3.45 | 0.85 | 0.57 | 3 | 1.94 | 1.42 | 0.85 | 2 | 1 | |
| 2.5 | 0.85 | 0.85 | 3 | 1.54 | 0.85 | 0.85 | 2 | No sill | |
| 3.05 | 0.85 | 0.57 | 2 | 1.86 | 0.85 | 0.57 | 2 | No sill | |
| 5.66 | 0.85 | 0.85 | 3 | 1.76 | 0.85 | 0.85 | 2 | 0 | |
| 6.67 | 0.85 | 0.57 | 3 | 2.17 | 1.42 | 0.85 | 3 | 0 | 4 |
| 4.4 | 0.85 | 0.57 | 2 | 1.7 | 1.42 | 0.57 | 2 | 0.5 | |
| 5.2 | 0.85 | 0.57 | 3 | 2.08 | 1.42 | 0.85 | 3 | 0.5 | |
| 2.7 | 0.85 | 0.85 | 3 | 1.59 | 0.85 | 0.85 | 2 | 1 | |
| 3.33 | 1.42 | 0.57 | 3 | 1.95 | 1.42 | 0.85 | 3 | 1 | |

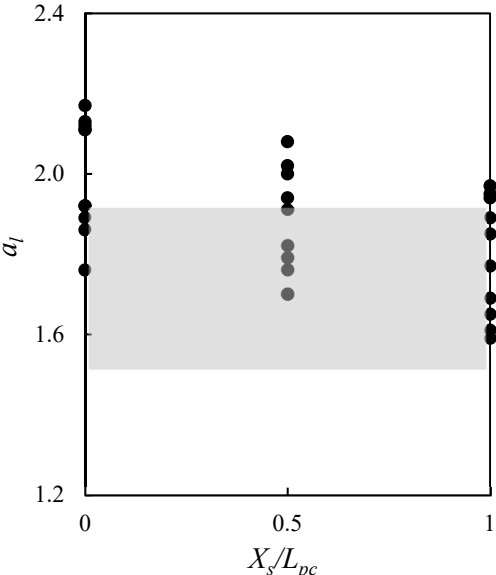
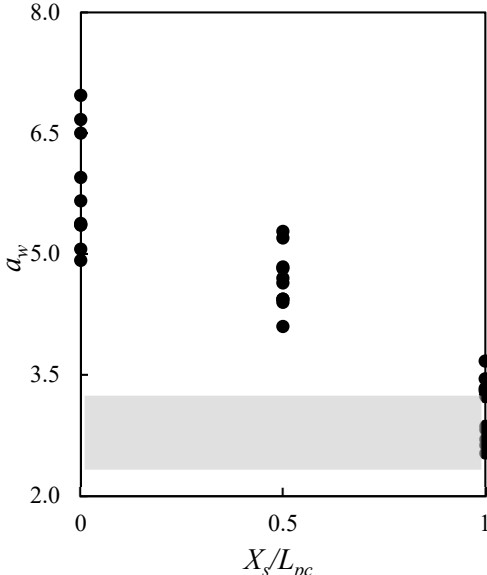

**Figure 11.** Dimensions of the hole as a function of the sill location. The grey rectangle corresponds to the range of values for the no-sill condition.

## 6. Discussion and Prospects for Further Research

The present experiments quantitatively demonstrated the efficiency of a transverse sill as a scour countermeasure for a complex pier with a specific geometry under a prescribed hydro-dynamic condition.

Sills are usually employed to stabilize riverbeds, in this case working as check dams. These structures impose a bed level at their location, thus setting a slope for an upstream reach (furthermore, it is well known that check dams are frequently built in series to restore river reach by imposing a milder slope in several sub-reaches upstream of each structure). Some scholars who attempted to mitigate local scour using a sill have also placed the latter downstream of the pier under investigation (Grimaldi et al., 2009a, 2009b [21,22]). The mechanism by which the downstream sill reduces the scour depth is hindering the transport of the sediment mobilized close to the pier uphill along the downstream slope of an erosion hole. Grimaldi et al. (2009a) [21] mentioned that the efficiency of the sill increased when the distance from the pier to the sill lowered, with a maximum scour reduction of around 25% when the sill was placed at zero distance from the pier (similar percentages of scour reduction were obtained by Tafarojnoruz et al., 2012) [37]. In addition, they spotted a delay in the sill effectiveness, because, in their experiments, the countermeasure became active when the scour hole reached the sill location, while the structure did not change the scour process in the early stages.

In this work, we detected the highest performance for an upstream sill. The latter was also used by Chiew and Lim (2003) [26], even if they placed it some distance upstream of the pier. An upstream sill starts working immediately (and it should be noted that, in the present study, the middle and downstream sills acted very soon, since, in any case, a countermeasure was placed at the pier). With the sill upstream of the pier, the scour dynamics become more complex, because the pier lies in a region prone to sill/apron scour. It is well known that a scour hole is expected at a sill (e.g., Marion et al., 2006 [38]; Ben Meftah and Mossa, 2006 [39], among others). This becomes more important when a downward step creates a water jet; however, the transition from a non-erodible to an erodible bed always induces some scour, even in the absence of an abrupt decrease in the bed elevation, due to local, small-scale variations in the sediment arrangement and bed surface. Therefore, using an upstream sill as a scour countermeasure creates a complex situation with an erosion pattern that results from the combination of sill scour, pier scour, and pier hiding. It was shown in this study that this combination is beneficial in

terms of the maximum scour depth, in agreement with previous studies of Saadati Pacheh Kenari et al. (2014) [29] for different geometries in a complex pier (inclined bridge pier group mounted on a solid rectangular foundation). The price of this is that of a wider scour hole because, if a sill occupies the entire width of the channel, some sill scour is expected along the entire width. Likely, this was not observed in the present experiment due to some sidewall effects that reduced the flow velocity close to the lateral walls. The secondary effects of a scour countermeasure, however, have yet to be thoroughly analyzed.

Finally, the present experiments enabled scour reductions of up to 30–40% of maximum scour depth to be obtained. The trend of the sill performance with $X_s/L_{pc}$ was quite stable, with the best efficiency for $X_s/L_{pc} = 0$. Some of the variability in the percentages was attributed to the geometric parameters of the piles and cap. Furthermore, when the cap was lowest, the effect of the foundation piles was reduced, as these components were hidden to the flow, with all the parameters instead contributing to the high elevation of the pile cap.

The scour values and reductions documented above are only related to the frontal piles, because they were those at which the maximum scour depth occurred. In the context of analyzing pier safety, the various scour values at the different piles should be also considered, as the pier stability may depend on a combination of the scour depth at different piles. However, in the present experiments, the sill induced a reduction in scour at all the piles, even with variable percentages. Therefore, the investigation of the scour depth at the frontal piles was considered, to detect a general increase in pier safety.

Since the investigated geometry is quite specific, there is some merit in asking ourselves whether the present results may be also representative of other configurations, and thus potentially of wider interest. Recently, Esmaeili Varaki et al. (2019) [11] found that the scour depth at the present pier could be estimated, albeit with some degree of uncertainty, with predictors developed for other geometries of complex piers (mostly with a vertical, circular column). Therefore, we expect that the scour reductions that were found in this study could also be applicable to other complex piers. Further experiments are needed to pass from expectation to certainty.

Moreover, the present results cannot be generalized to other hydro-dynamic and countermeasure conditions, since a single combination of flow rate and water depth was used in this work. In particular, there is some interest in moving towards conditions of higher velocity, for two main reasons. First, in case the threshold velocity for sediment transport in the undisturbed reach was underestimated, increasing the velocity would lead to measure larger scour depths (close to the threshold conditions). Second, the proposed countermeasure would also need to be tested under live-bed conditions, where the elevation of the river bed continuously changes. In this respect, the results of Wang et al. (2018) [30] and Yang et al. (2018 and 2019) [9,10] are encouraging, but thorough verification is still needed. Furthermore, conditions with a skewed flow could be also considered, as many bridges are skewed due to crossing geometries.

Finally, the present experiments were related to clear-water scour in a stable bed and clean water. Other effects may need to be carefully accounted for before the use of a bed sill upstream of a pier is proposed. For example, in a degrading bed, the local scour process at the system of pier and sill will be coupled with the general scour due to bed degradation, and the possibility for the sill to hinder both bed degradation and local scour needs to be ascertained. Second, in the presence of debris material, the performance of any scour countermeasure could significantly change, and a sill is also prone to this shortcoming (e.g., Tafarojnoruz et al., 2012 [37]). These effects need to be thoroughly investigated in follow-up campaigns.

Finally, the transferability of any conceived countermeasure to the real world is not a trivial issue due to construction issues. The present manuscript has dealt with an idealized experimentation where everything is possible, as many other studies on scour countermeasures are available in the literature. In practice, constructing a sill close to an existing bridge would require a hole close to the piers to be dug, as also occurs for riprap installation. The need to create a hole, although limited to the construction term, is an issue that needs to be

carefully considered to avoid undermining undesired structures. A sill construction should be thought of only for bridges with deep foundations, as it might be impossible for bridges with direct foundations.

## 7. Conclusions

Based on the results of the experiments performed in the present study, a transverse sill may be a suitable countermeasure for reductions in local scour at a studied complex pier.

When placed at the downstream edge of the pier, a sill reduced the maximum scour depth (limitedly, by around 10%), hindering the motion of the eroded sediment out of the scour hole. The shape of the latter was not significantly different from that measured for an unprotected pier. By contrast, when a transverse sill was placed at the upstream edge of a pier, it might reduce the maximum scour depth by up to 30–40%, because it blocks the flow that impacts the bottom components of the pier. Among those in this study, the upstream sill placement was the most effective, even if the relative length and width of the scour hole increased compared to those of an unprotected pier (with the increase in relative width becoming higher than the increase in relative length).

Other dimensionless parameters (those related to cap elevation and thickness, as well as pile number and diameter) have an impact on the scour reduction that one may achieve by installing a bed sill; thus, ad hoc studies may be necessary to assess the countermeasure performance for complex piers with a specific geometry. However, the effect of these other parameters was, in the present experiments, more limited than that of the sill placement.

**Author Contributions:** Conceptualization, M.E.V., N.T. and A.R.; methodology, M.E.V., N.T. and A.R.; software, M.E.V., N.T. and A.R.; validation, M.E.V., N.T. and A.R.; formal analysis, M.E.V., N.T. and A.R.; investigation, M.E.V., N.T. and A.R.; resources, M.E.V., N.T. and A.R.; data curation, M.E.V., N.T. and A.R.; writing—original draft preparation, M.E.V., N.T. and A.R.; writing—review and editing, M.E.V. and A.R.; visualization, M.E.V., N.T. and A.R.; supervision, M.E.V. and A.R.; project administration, M.E.V., N.T. and A.R.; funding acquisition, M.E.V., N.T. and A.R. All authors have read and agreed to the published version of the manuscript.

**Funding:** This research received no external funding.

**Institutional Review Board Statement:** Not applicable.

**Informed Consent Statement:** Not applicable.

**Data Availability Statement:** All data, models, and code generated or used during the study appear in the submitted article.

**Conflicts of Interest:** The authors declare no conflict of interest.

## Symbols

| | |
|---|---|
| $B$ | flume width |
| $D$ | column width |
| $D_{50}$, $\sigma_g$ and $\rho_s$ | median size, uniformity coefficient and density of sediment |
| $d_p$ | diameter of piles |
| $D_{pc}$ and $L_{pc}$ | transverse and stream-wise dimensions of the pile cap |
| $d_s$ | the time-dependent maximum scour depth |
| $d_{s5}$ | the maximum scour depth at $tU/D = 1.5 \times 10^5$ |
| $d_{sf}$ | the maximum scour depth measured after 25 h |
| $g$ | acceleration due to gravity |
| $m$ | number of piles in the stream-wise direction |
| $n$ | number of piles in the transverse direction |
| $t$ | time |
| $T_{pc}$ | thickness of the pile cap |
| $U$ | section-averaged flow velocity |
| $X_s$ | stream-wise coordinate of the sill, measured from the upstream edge of the pile cap |
| $y$ | flow depth |

| $Z$ | elevation of the pile cap above the non-scoured bed elevation |
| $Z_s$ | elevation of the sill above the non-scoured bed level |
| $\alpha$ | pier inclination |
| $\rho$ and $\mu$ | density and viscosity of water |

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
