# Peer review of "Using a Bed Sill as a Countermeasure for Clear-Water Scour at a Complex Pier with Inclined Columns Footed on Capped Piles"

_hydrology, doi:10.3390/hydrology9040065_

Round 1
Reviewer 1 Report
Please revise the manuscript based on the attached file.

Author Response
Comments by Reviewer 1
The reviewed paper presents an experimental study on application of bed sill as scour countermeasure around a complex pier. The investigation is interesting and may be published as a new article provided that the authors address all the following comments and revise the manuscript accordingly.
Reply: We thank the Reviewer for her/his interest in the paper. We have revised the paper based on the comments received, that have been either accepted or rebutted. In this letter we provide point-by-point replies.
- Page 2 (last paragraph): It is indicated that column inclination increases the scour depth. what is the reason in view point of horseshoe vortex and wake vortices? Please mention in the paper.
Reply: As shown in following plot (taken from an earlier study of Esmaeili Varaki and Saadati Pachekenari, 2016), with an inclination angle of 28° the scour depth was maximum. Further increase of the inclination angle up to 38° reduced the maximum scour depth due to reduction of stream line diversion to the bed and the formed horseshoe vortex and wake vortices. Also, increase of the inclination angle from 38° to 45°, slightly increased the maximum scour depth because the diverted stream line velocity at near the bed and the corresponding formed horseshoe vortex increases.
Esmaeili Varaki, M.; Saadati Pachekenari, S.S. Investigating experimentally the effect of installation of piers group on foundation on scour depth around bridge piers. Journal of water and soil, 2016, 25(4): 27-39.
- Page 3 (last line): It is stated that a bed sill downstream of a pier reduces scour hole dimension. First, along longitudinal or transverse direction? Second, as maximum scour depth around bridge pier is the most important parameter in pier stability, how effective is a bed sill as scour countermeasure around piers? please explain within the paper.
Reply: We have added quantitative information to the Introduction.
- Page 4 (second paragraph): Regarding the sentence "....to use a transverse sill as a scour countermeasure...", is it possible to implement a sill in any other direction?
Reply: We have not investigated that, honestly. In general, bridge crossings may be skewed (that is what stimulated extensive research on pier scour with skewed flow) and we guess that a sill could also be, but we have no information about quantitative performance indicators.
- Page 4: It seems that two inclined columns were used in your tests (Figure 3). Do you think that number of inclined columns is an effective parameter to be included in Eq. (1)?
Reply: In a general view, one should include any possible parameter and, therefore, the column number should not be excluded a priori. However, we have used this bridge geometry to mimic the bridge of Ahvaz mentioned in the paper.
- Page 5: Is Eq. (3) in accordance with the previous studies? If yes, please add a reference for it.
Reply: This equation is (standard but) original development for the present study.
- Page 6: I see a ‘dip’ in velocity profile S2. This phenomenon occurs in a narrow channel condition when the ratio of flow depth/channel width is relatively large or flow is not fully developed. How did you measure this profile (a Pitot tube or ADV)? Can you perform further analysis of that profile to ensure flow was fully developed at pier location?
Reply: The S1and S2 are located 0.5 and 1.5 m upstream of the pier, respectively. The dip phenomenon did not occur at the S1 profile. Velocity measurements have been performed using an ADV with down-looking probe. After this experimental campaign the flume configuration was changed to conduct a new program and it is unfortunately impossible to perform new measurement.
- Page 6: How did you choose 28 degrees as pier inclination? Did you consider a prototype pier? Please notify in the paper.
Reply: Yes. The pier geometry studied in the present manuscript was stimulated by the 8th bridge of Ahvaz on the Karun River, Iran (this bridge is mentioned in the introduction and depicted in figure 1 of the manuscript).
- Page 7: why did you choose 25 hours as test duration? In general, it takes several days to obtain equilibrium condition.
Reply: Some preliminary scour experiments were performed for a duration of 72 h; the results of these experiments (not shown here) indicated that at a dimensionless time tU/D = 1.5×105 (corresponding to 7 h) a scour depth as large as 70–80% of the equilibrium one could be reached, while at tU/D = 5.4×105 (25 h) the scour depth was more than 90% of the equilibrium one. In the revised manuscript we have added this information to the section on experimental procedures.
- I found the following papers about scour around a complex pier and combination of a thick bed sill with a collar as scour countermeasure around a uniform pier under steady and unsteady conditions. I suggest the authors to use these articles in their study to notify the importance of this subject:
- Effects of pile cap thickness on the maximum scour depth at a complex pier
- Combined flow-altering countermeasures against bridge pier scour
- Bridge pier scour mitigation under steady and unsteady flow conditions
Reply: The mentioned references were added.
- Page 11 (last paragraph): what about flow alignment? Do you have any idea about the effectiveness of a sill in oblique flow?
Reply: As mentioned in the reply to comment 3, this was not investigated.
- Page 13: which instrument did you use to acquire scour contour? What about its accuracy?
Reply: The topography of the scour hole was surveyed using a laser distance sensor with an accuracy of ±1 mm. This is mentioned in the section on experimental procedures. The measuring grids for the surveys was 2×2 cm.
- Figure 10(c): I observe strong variation of contours, in particular for 0 and -0.5. Can you justify it?
Reply: These irregularities occur in the highest part of the scour hole where small changes of the bed elevation (as the Reviewer pointed out we are in a range of few mm) may induce large changes in the profile of the hole edge. It is probably an effect related to interpolation of measured scour depths.
- What is the unit of contours in Figure 10? please mention in the paper. Maybe cm?
Reply: The unit is cm. We have mentioned this in the caption of Fig. 10.
- Do the authors think that construction of a bed sill close to an available pier may affect the stability of that pier? Is it practical?
Reply: The Reviewer raised a crucial issue of transferability of any invented countermeasure to the real world. Very important indeed. We guess that demonstrating that a countermeasure would work is a first step, to be investigated by hydraulic engineers. Construction is to be handled by structural and geotechnical engineers and, in this sense, our experience is limited. However, realizing a sill would require digging a hole, an operation that is commonly done for, for example, placing riprap (that is a diffuse intervention). We think that applicability of any countermeasure that requires digging a hole would be possible only for pile foundations and not for direct ones. We have added a mention to this issue in the Discussion section.

Reviewer 2 Report
This work can be accepted for publication in MDPI journals.
Author Response
Comments by the Academic Editor
- The manuscript presents using a bed sill as a countermeasure for clear-water scour at a complex pier with inclined columns footed on capped piles, which is interesting. The subject addressed is within the scope of the journal.
- However, the manuscript, in its present form, contains several weaknesses. Appropriate revisions to the following points should be undertaken in order to justify recommendation for publication.
Reply: We have addressed the comments and revised the manuscript as explained below.
- For readers to quickly catch your contribution, it would be better to highlight major difficulties and challenges, and your original achievements to overcome them, in a clearer way in abstract and introduction.
Reply: The abstract has been revised at multiple instances, highlighting that a complex pier geometry increases the complexity of the flow field, that this is the first time a sill was tested as a scour countermeasure for the used pier geometry, and that the experimental campaign was wide.
We have also extensively revised the introduction, that had a tortuous structure. The revised section should better convey the motivation and objectives of the study.
- It is shown in the reference list that the authors have several publications in this field. This raises some concerns regarding the potential overlap with their previous works. The authors should explicitly state the novel contribution of this work, the similarities, and the differences of this work with their previous publications.
Reply: As mentioned in the introduction, the previous works have investigated a complex pier with a rectangular foundation. The present work relates to a complex pier with a cap footed on an array of piles. In this geometry the flow can pass through the spaces between the piles resulting in a different scour pattern. It is mentioned, in both abstract and introduction, that this geometry has never been studied before in combination with a scour countermeasure.
- p.1 - a transverse bed sill is adopted as a countermeasure for local scour at a bridge pier. What are other feasible alternatives? What are the advantages of adopting this approach over others in this case? How will this affect the results? The authors should provide more details on this.
Reply: Several other countermeasures are mentioned in the manuscript introduction. In principle, one could use any of them to protect an existing bridge, after appropriate design of the mitigation structure. In this manuscript we intended to perform a prototypal study of the performance of a sill, after encouraging results have been obtained for other pier geometries. Answering the question “what are feasible alternatives?” would mean include in the manuscript a textbook review of scour countermeasures, but we think that this is beyond the scope of the paper.
- p.1 - an array of piles, a pile cap and two inclined columns with rectangular section above the cap are adopted in the experiments. What are the other feasible alternatives? What are the advantages of adopting this compound over others in this case? How will this affect the results? More details should be furnished.
Reply: A complex pier may have any geometry. Below there is a random pick of images from the Internet. So, “other feasible alternatives” are infinite and, in this sense, the questions cannot be answered. We have used the present geometry, as mentioned in the abstract and introduction, because it is representative of an existing bridge.
- p.5 - the flume layout as shown in Fig. 4 is adopted in this study. What are other feasible alternatives? What are the advantages of adopting this layout over others in this case? How will this affect the results? The authors should provide more details on this.
- p.5 - uniform sand with a particle size of 0.7 mm is adopted as the sediment. What are other feasible alternatives? What are the advantages of adopting this sediment over others in this case? How will this affect the results? The authors should provide more details on this.
- p.6 - a specific bridge pier model is adopted in the experiments. What are other feasible alternatives? What are the advantages of adopting this model over others in this case? How will this affect the results? The authors should provide more details on this.
Reply to 7: Generally, the investigation of local scour around bridge piers has been conducted in experimental flumes like the one used in the present manuscript. Of course variations are possible. For the bed: one can used a recess section with the bed fixed upstream; this fixed bed can be realized by larger particles, like in the present case, or by the same particles glued onto the flume bottom; or, one can use an erodible bed along the entire flume. For the inlet: one may use different kinds of flow straighteners or not. For the outlet: one can use different devices for tailgate regulation. Again, answering the question “what are feasible alternatives?” is impossible without re-writing a textbook on experimental hydraulics for open-channel flows.
Reply to 8: Similarly, the alternatives for sediment are almost infinite and relatively obvious. One can use natural or artificial particles, change the median particle size, use quasi-uniform or strongly non-uniform granulometry… With due respect to the Editor, we think that a scholar needs to show and discuss what has been done rather than presenting a never-ending list of what could have been done instead.
Reply to 9: Same as that to comment 6.
- p.6 - “…Constriction effects were excluded because the.…” More justification should be furnished on this issue.
Reply: the values of the ratios between the flume width and the sizes of the pier elements have been added to the revised manuscript, for a direct comparison to a threshold value provided by the literature.
- p.6 - a single hydrodynamic condition of y = 0.225 m and U = 0.26 m/s is adopted in the experiments. What are other feasible alternatives? What are the advantages of adopting this hydrodynamic condition over others in this case? How will this affect the results? The authors should provide more details on this.
Reply: Exploring more hydro-dynamic conditions (or maybe an infinite range of hydro-dynamic conditions) would have returned more general results, obviously. It is acknowledged in the manuscript (both in abstract and discussion) that the results presented here cannot be extended to other conditions, but are an encouraging basis for follow-up research.
- p.8 - the experimental runs as shown in Table 1 are adopted in this study. What are the other feasible alternatives? What are the advantages of adopting these parameters over others in this case? How will this affect the results? More details should be furnished.
Reply: The experimental campaign was organized with a systematic variation of a set of control parameters. Again, one could have chosen other control parameters, or test along much wider ranges of variation. As mentioned in the paper, the present experimental campaign comprehended more than 140 runs, thus requiring a great deal of effort to be put in experiment realization.
- p.13 - “…most of the plots show an increase of the sill performance as Z/D increased, as the circle is frequently the lowest symbol in a column while the star is frequently the highest; this effect is reasonable, because.…” More justification should be furnished on this issue.
Reply: According to the performance of sills on reduction of scour depth, upstream sill had the most effect on the reduction of scour depth. For Z/D=0, The change in pile cap thickness and pile diameter had a similar effect on the sills performance and the effect of change in pile cap thickness in relative pile diameter of 0.85 was more noticeable for the middle and downstream sills. The effect of changing the relative diameter of the pile was also more noticeable for the middle and downstream sills in the relative pile cap thickness of 1.42. For Z/D=1, change in the pile cap thickness had the most effect on sills performance. In general, increasing the pile cap thickness caused reduction of the sills performance. Increasing the pile diameter resulted in increase of the sills performance except in case of Tpc/D=0.85 and N=2, which the performance of the upstream and downstream sills decreased with increasing the pile diameter. The increase in number of piles from 2 to 3, had the least effect on sills efficiency.
For Z/D=2, change on pile diameter had the most effect on sills performance and the increase of the number of piles from 2 to 3, had the least effect on sills efficiency. In this level Increasing the pile diameter caused increase of the sills performance except for Tpc/D=0.85 and N=3, which Increasing the pile diameter caused to reduce of the sills performance. By increasing the piles number, the sills performance fluctuated. For dp/D=0.85 by Increasing the pile cap thickness, sills performance increased and for Tpc/D=0.85 and N=3 decreased and also for Tpc/D=0.85 and N=2 the performance of the upstream and downstream sills decreased with increasing the pile diameter. For Z/D=3, change of pile cap thickness had the most effect on sills performance and increase of the number of piles from 2 to 3, had the least effect on sills efficiency. Comparison of results showed that increasing the pile diameter led to decrease the upstream sills performance except for case Tpc/D=1.42 and N=2. In this level for dpc/D=0.85, increasing the pile cap thickness causes to decrease in the sills performance and for dpc/D=1.42, increase of the pile cap thickness led to increase of the sills performance.
For Z/D=4, change of pile cap thickness had the most effect on the sills performance and increase of the number of piles from 2 to 3, had the least effect on sills efficiency. In this level, Increasing the pile cap thickness caused to increase of the sills performance for all cases and also by increasing the pile diameter for Tpc/D=1.42, the sill performance increased. At this level, effect of increasing the pile number on performance of sills was negligible.
- p.16 - “…The relationship between the transverse sill and the bed in an upstream reach was possibly the reason for which.…” More justification should be furnished on this issue.
Reply: Even if we think that our arguments were reasonable, in the revised manuscript we have removed this material that could be considered too speculative.
- Some key parameters are not mentioned. The rationale on the choice of the particular set of parameters should be explained with more details. Have the authors experimented with other sets of values? What are the sensitivities of these parameters on the results?
Reply: The response is similar to that to comment 12. We have investigated the effect of the pile cap thickness, the pile diameter and the cap elevation on the sill performance in reducing the maximum scour depth. If we had experimented other sets of values, they would have been included in the manuscript. We obviously cannot say which effect other parameters (which ones? One can hardly imagine what “some key parameters” means…) could have had if we have not investigated them.
- Some assumptions are stated in various sections. Justifications should be provided on these assumptions. Evaluation on how they will affect the results should be made.
Reply: The assumptions made in the manuscript are rather standard for scour investigations. To avoid scale effects, we chose appropriate dimensions for bridge pier, sediment size, depth and mean velocity of flow.
- The discussion section in the present form is relatively weak and should be strengthened with more details and justifications.
Reply: The Discussion section contains a short summary of the achievements, comparison with prior studies where a sill was employed, critical issues (one of which has been added based on a comment of Reviewer 1) and prospects for further research. We believe that this section can stimulate the thought of readers. Honestly we were unable to catch the weak points that the Editor had in mind, since her/his comment did not include specific issues.
- Moreover, the manuscript could be substantially improved by relying and citing more on recent literatures about contemporary real-life case studies of experiments and modeling on hydraulic structures such as the followings:
l El-Mahdy, M.E.S., et al., “Experimental Method to Predict Scour Characteristics Downstream of Stepped Spillway Equipped With V-Notch End Sill,” Alexandria Engineering Journal 60 (5): 4337-4346 2021.
l Khalifehei, K., et al., “Experimental Modeling and Evaluation Sediment Scouring in Riverbeds around Downstream in Flip Buckets,” International Journal of Engineering 33 (10): 1904-1916 2020.
l Sun, X.P., et al., “Hybrid model of support vector regression and fruitfly optimization algorithm for predicting ski-jump spillway scour geometry,” Engineering Applications of Computational Fluid Mechanics 15 (1): 272-291 2021.
Reply: Thanks for mentioning these papers. We are sure that they contain good science and will stimulate our thoughts in following research. However, a huge literature exists in relation to scour and we think that we should focus on scour around bridge piers in our reference list.
- In the conclusion section, some recommendations are made for further investigation. Why are they not performed in this study? More justifications should be furnished on this.
Reply: As mentioned, we are reporting more than 140 runs in this manuscript. This number is relatively large compared to those in other articles on pier scour and the experimental campaign took several months. As in many research areas, one has to progress step by step because it is impossible to treat all relevant issues in a single paper.

Round 2
Reviewer 1 Report
The manuscript may be published in present form.
Author Response
We thank the Editor and Reviewers for their interest in our paper. From 5 references that we are authors, we removed one of the self-references which I am the first author. We deeply used the other references in the text of the paper and it is not possible to remove those. Furthermore, some minor revision was made. We hope to get acceptance after this round of revision.